# Decomposing multimodal embedding spaces with group-sparse autoencoders

**Chiraag Kaushik**
School of Electrical and Computer Engineering
Georgia Institute of Technology
Atlanta, GA 30308, USA
ckaushik7@gatech.edu

**Davis Barch**
Dolby Laboratories
San Francisco, CA 94103
davis.barch@dolby.com

**Andrea Fanelli**
Dolby Laboratories
San Francisco, CA 94103
andrea.fanelli@dolby.com

## Abstract

The Linear Representation Hypothesis asserts that the embeddings learned by neural networks can be understood as linear combinations of features corresponding to high-level concepts. Based on this ansatz, sparse autoencoders (SAEs) have recently become a popular method for decomposing embeddings into a sparse combination of linear directions, which have been shown empirically to often correspond to human-interpretable semantics. However, recent attempts to apply SAEs to multimodal embedding spaces (such as the popular CLIP embeddings for image/text data) have found that SAEs often learn "split dictionaries," where most of the learned sparse features are essentially unimodal, active only for data of a single modality. In this work, we study how to effectively adapt SAEs for the setting of multimodal embeddings while ensuring multimodal alignment. We first argue that the existence of a split dictionary decomposition on an aligned embedding space implies the existence of a non-split dictionary with improved modality alignment. Then, we propose a new SAE-based approach to multimodal embedding decomposition using cross-modal random masking and group-sparse regularization. We apply our method to popular embeddings for image/text (CLIP) and audio/text (CLAP) data and show that, compared to standard SAEs, our approach learns a more multimodal dictionary while reducing the number of dead neurons and improving feature semanticity. We finally demonstrate how this improvement in alignment of concepts between modalities can enable improvements in the interpretability and control of cross-modal tasks.

## 1 Introduction

Multimodal representation learning models like CLIP (Radford et al., 2021), SigLIP (Zhai et al., 2023), and CLAP (Elizalde et al., 2023; Wu et al., 2023) aim to learn a shared latent representation for various types of input data, such as images, text, and audio. The goal of these multimodal encoders is to learn embeddings which are in some sense (e.g., cosine similarity) "similar" to one another when the input data contains shared semantic information, regardless of the data's modality. The ability of such models to learn expressive multimodal data representations has been extensively demonstrated by their excellent performance on cross-modal tasks like zero-shot classification (Radford et al., 2021), retrieval (Elizalde et al., 2023; Wu et al., 2023), and generation (Ramesh et al., 2022). However, despite the promising empirical performance of these models, the precise way in which they encode semantic information is not well-understood.

Motivated primarily by the task of interpreting the behavior of large language models, sparse autoencoders (SAEs) have recently become a popular approach for understanding how human-interpretable concepts are encoded in the embeddings of modern neural networks. Building on classical ideas

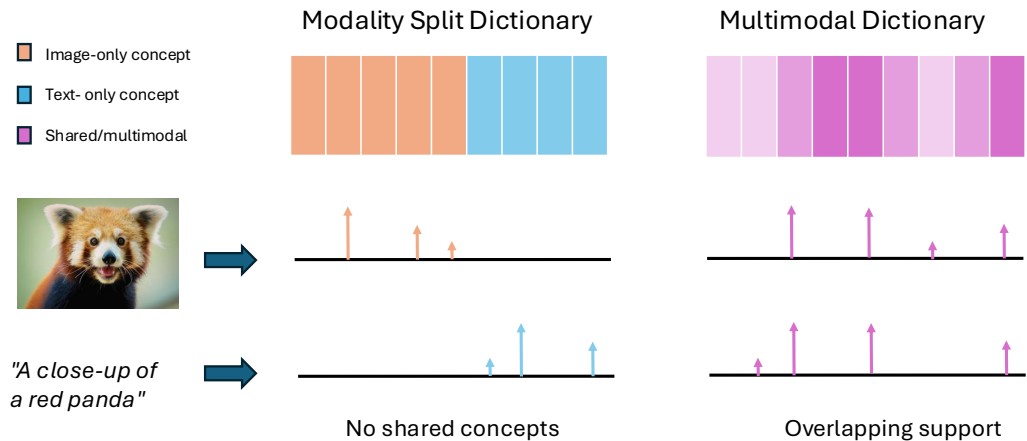

Figure 1: **Split versus multimodal dictionaries:** (Left) Standard SAEs trained on aligned embeddings from different modalities (like CLIP) tend to learn dictionary vectors ("concepts") which activate only for input embeddings of one modality. (Right) We develop an SAE-based approach for learning dictionaries where multimodal embeddings containing the same semantic information have similar sparse dictionary decompositions.

from dictionary learning and sparse coding, SAEs are unsupervised models that aim to decompose neural network embeddings as sparse linear combinations of a large (overcomplete) set of dictionary vectors. It has been empirically observed in a variety of domains that these dictionary elements tend to correspond to more human-understandable concepts, allowing for improvements in interpretability, steering, and control of models through linear algebraic manipulation of embeddings (Cunningham et al., 2023; Gao et al., 2024; Pach et al., 2025). The so-called "Linear Representation Hypothesis" proposes the existence of such a linear semantic decomposition (Elhage et al., 2022; Park et al., 2023), and this notion has been empirically supported for various concepts such as query refusal in language models (Arditi et al., 2024) and the presence of certain facial characteristics in images (Voynov & Babenko, 2020).

Recent attempts to apply sparse autoencoders to the aligned embedding space of multimodal models like CLIP have obtained mixed results. On the one hand, the neurons learned by SAEs trained on CLIP do demonstrate improved monosemanticity and often correspond to human-interpretable concepts (Yan et al., 2025; Pach et al., 2025). However, many works identify a *"split-dictionary"* *phenomenon*, where many of the sparse features are only ever active for inputs coming from a single modality (Papadimitriou et al., 2025; Costa et al., 2025). We depict this pictorially in Figure 1. In other words, SAEs learn to map embeddings which are *aligned* between modalities to sparse features which may be more semantic, but which have very poor modality alignment. This phenomenon limits the potential of SAEs in cross-modal tasks like retrieval or generation, where, for example, manipulation of text embeddings based on a certain "concept" should ideally be able to influence a visual or audio output.

In this paper, we focus on answering the question of how to mitigate the tradeoff between monosemanticity and modality alignment in SAEs trained on multimodal embeddings. We investigate whether it is possible to learn multimodal linear concept dictionaries and we design a model which mitigates the implicit bias towards split-dictionaries observed in regular SAEs. We make the following contributions:

1. **New metrics for multimodal SAEs.** Motivated by the image-specific score proposed in (Pach et al., 2025), we formally define a paired-modality monosemanticity score which quantifies the semanticity of a neuron with respect to any two modalities, taking a value of $0$ if the neuron never co-activates for inputs of both modalities and with higher value if the co-activating inputs contain similar semantic information. Our metrics generalize existing methods for measuring semanticity at the neuron-specific level to the setting where SAE neurons are expected to co-activate for data from diverse modalities.

2. **Argument for the existence of multimodal dictionaries.** Under realistic technical assumptions, we prove that the existence of a split dictionary on an aligned embedding space implies the existence of a non-split dictionary with strictly improved modality alignment. This suggests that the poor modality alignment in SAEs is not exclusively due to limitations of the LRH, but rather is also a product of the implicit bias of SAEs trained exclusively on reconstruction loss.

3. **Group-sparse autoencoder architecture with improved multimodality.** We propose the use of a group-sparse loss function acting on paired samples during the training of SAEs, along with a form of cross-modal random masking, which together encourage an SAE to learn multimodal dictionary vectors. We train the proposed model on image/text (CLIP) and audio/text (CLAP) data, and evaluate our approach using a variety of metrics. Our results show improvements in the number of multimodal concepts, a reduction in the number of dead neurons, improvements in zero-shot cross-modal tasks, and improved capabilities for interpreting multimodal models (compared to regular SAEs). We also note that, to our knowledge, joint embedding spaces for audio/text data have not been analyzed using SAEs in prior literature.

## 2   RELATED WORK

**Dictionary learning and sparse coding**   The decomposition of a data sample $x \in \mathbb{R}^d$ as a sparse linear combination of learned dictionary elements $W \in \mathbb{R}^{d \times p}, p \gg d$, is a well-studied problem in many applications, such as image processing (Olshausen et al., 2009), medical imaging (Mailhé et al., 2009), and audio processing (Plumbley et al., 2009). Several popular algorithms like the Method of Optimal Directions (MOD) (Engan et al., 1999), the K-SVD (Aharon et al., 2006), and unrolled ISTA (Gregor & LeCun, 2010) have been proposed for the joint learning of sparse codes and the dictionary. A few works have considered modifications of these algorithms with additional structural constraints on the sparse codes, typically by adding a sparsity-promoting norm penalty such as the group norm or $L_{2,1}$ norm (Bengio et al., 2009; Li et al., 2012). Compared to these works, we also apply a group-sparsity penalty to the loss function of a sparse dictionary learning method (the SAE), but in our setting, the groups are given by pairs of data from different modalities containing the same concept. We additionally incorporate a type of per-group random masking to promote shared sparsity structure within each group, and we show that the combination of these two regularization techniques is effective in the context of multimodal embedding spaces.

**SAEs and neural network interpretability**   Sparse decompositions of neural network embeddings have recently been studied in many works in the field of neural network interpretability (El-hage et al., 2022; Cunningham et al., 2023; Rajamanoharan et al., 2024). Many of these works leverage (variants of) sparse autoencoders (SAEs), which are simple neural network-based models for sparse dictionary learning. In several of these works, the dictionary elements learned by sparse autoencoders (SAEs) trained on embeddings from language models have been empirically shown to often correspond to interpretable high-level concepts (Bussmann et al., 2024; Gao et al., 2024). Inspired by the efficacy of SAEs in language models, recent works have also proposed the use of SAEs in other domains, such as for vision models (Thasarathan et al., 2025; Fel et al., 2025) and neural data (Freeman et al., 2025).

**Decomposition of multimodal embeddings**   While most recent works on SAEs have studied interpretability of language or vision models, a few recent works also have applied SAEs to modality-aligned representations, such as those from models like CLIP, SigLIP, and DeCLIP (Papadimitriou et al., 2025; Pach et al., 2025; Yan et al., 2025; Zaigrajew et al., 2025). Most of these works train SAEs to reconstruct embeddings originating from visual and text data, using the same architectures as are prevalent in the language model setting. Several works that train in this way observe that many SAE neurons activate only for inputs of one modality (Costa et al., 2025; Bhalla et al., 2024b; Papadimitriou et al., 2025; Yan et al., 2025). Previous works have addressed this by explicitly learning transformations from one modality to the other or by identifying co-activating pairs of dictionary elements (Papadimitriou et al., 2025). Unlike these approaches, we seek to identify a first-principles approach to learning a multimodal dictionary using SAEs, overcoming the bias of vanilla SAEs towards modality-split dictionary elements. The authors of (Costa et al., 2025) also cite this discrepancy and find that an SAE variant based on the classical matched-pursuit algorithm can help increase

the number of multimodal concepts, but their focus is primarily on learning hierarchical concepts, rather than explicitly mitigating the split-dictionary phenomenon.

## 3  MODALITY ALIGNMENT AND SEMANTICITY IN SAEs

In this section, we provide an overview of the sparse autoencoder (SAE) model, formally define modality-split dictionaries, and describe a proposed metric for quantifying the monosemanticity of multimodal SAEs.

### 3.1  SPARSE AUTOENCODERS AND THE SPLIT DICTIONARY PHENOMENON

We consider the general setting where a multimodal encoder maps data from multiple modalities into a joint representation space $\mathcal{X} \subseteq \mathbb{R}^d$. Motivated by the hypothesis that neural network representations consist of linear superpositions of interpretable "concepts" (Elhage et al., 2022), dictionary learning aims to learn an overcomplete dictionary $\boldsymbol{W} \in \mathbb{R}^{d \times p}$ such that representations $\boldsymbol{x} \in \mathcal{X}$ can be decomposed as $\boldsymbol{x} \approx \boldsymbol{W}\boldsymbol{z}$, for some sparse vector $\boldsymbol{z} \in \mathbb{R}^p$.

**Sparse autoencoders (SAEs)** are one popular method for learning this type of dictionary decomposition on neural network embeddings (Cunningham et al., 2023). Given an input $\boldsymbol{x} \in \mathbb{R}^d$, SAEs take the following form:

$$\boldsymbol{z} = \boldsymbol{\Pi}(\boldsymbol{W}_{enc}(\boldsymbol{x} - \boldsymbol{b}_0) + \boldsymbol{b})$$
$$\hat{\boldsymbol{x}} = \boldsymbol{W}_{dec}\boldsymbol{z} + \boldsymbol{b}_0.$$

Here, $\boldsymbol{b}_0 \in \mathbb{R}^d$ is a bias applied prior to encoding/decoding, $\boldsymbol{W}_{enc} \in \mathbb{R}^{p \times d}$ is the encoder matrix, $\boldsymbol{b}$ is the encoder bias parameter, and $\boldsymbol{W} \in \mathbb{R}^{d \times p}$ is the decoder matrix. The projection function $\boldsymbol{\Pi} \colon \mathbb{R}^p \to \mathbb{R}^p$ is chosen to promote sparsity in the latent vector $\boldsymbol{z}$. In this paper, we focus our attention on the TopK function (Gao et al., 2024), which keeps only the $K$ largest elements of the input and sets the remaining elements to $0$. However, we note that our proposed methodology can in principle be applied to other common SAE variants. Other common choices of $\boldsymbol{\Pi}$ in recent literature include the ReLU (Cunningham et al., 2023), BatchTopK (Bussmann et al., 2024), and JumpReLU (Rajamanoharan et al., 2024) functions, and these variants are often trained with additional $\ell_1$ regularization to promote sparsity of $\boldsymbol{z}$. In the following, we refer to the decoder matrix $\boldsymbol{W}_{dec}$ as the *dictionary matrix* and its columns as the *dictionary vectors*, or *concept vectors*. We will refer to $\boldsymbol{z}$ as the *SAE latent vector* or *sparse code*, and its coordinates as *neurons* of the SAE. The SAE parameters are learned by optimizing the reconstruction error $\|\boldsymbol{x} - \hat{\boldsymbol{x}}\|_2^2$ over all training samples $\boldsymbol{x}$ in the training set $\mathcal{D}$.

Recent attempts to train classical SAEs on multimodal embeddings from models like CLIP and SigLIP have found that the learned dictionary elements tend to correspond primarily to only one of the input modalities (Papadimitriou et al., 2025; Costa et al., 2025). In other words, many of the features in the SAE's latent space tend to only be active for inputs corresponding to a single modality. We define this property of a dictionary formally below:

**Definition 1** (Modality-split dictionary). *Let $\boldsymbol{W} \in \mathbb{R}^{d \times p}$ with $p \gg d$ be a dictionary matrix that admits a $K$-sparse decomposition of any $\boldsymbol{x} \in \mathcal{X}$. Then, $\boldsymbol{W}$ is said to be **modality-split** with respect to $\mathcal{X}$ if, for any two embeddings $\boldsymbol{x}^{(1)}, \boldsymbol{x}^{(2)} \in \mathcal{X}$ corresponding to data from different modalities,*

$$supp(\boldsymbol{z}^{(1)}) \cap supp(\boldsymbol{z}^{(2)}) = \emptyset,$$

*where $\boldsymbol{x}^{(1)} = \boldsymbol{W}\boldsymbol{z}^{(1)}$ and $\boldsymbol{x}^{(2)} = \boldsymbol{W}\boldsymbol{z}^{(2)}$ are any $K$-sparse decompositions of $\boldsymbol{x}^{(1)}$ and $\boldsymbol{x}^{(2)}$ in the dictionary.*

An important consequence of this definition is that *paired* data samples of different modalities (such as an image and a textual description of the same image's content) have sparse codes with no overlapping support, even though the original embedding space is aligned between modalities. In other words, classical SAEs trained on popular *aligned* multimodal embedding spaces have *poor modality alignment* in the SAE's latent space. A crucial application of SAE embedding decompositions is the steering or manipulation of SAE latents for interpretability and control of downstream tasks (Bhalla et al., 2024b). In cross-modal applications, therefore, it is important for data from different modalities that contain similar semantic information to have aligned sparse codes. The remainder of the paper is dedicated to understanding when and how this can be achieved in SAE-based models.

## 3.2 MEASURING SEMANTICITY ACROSS MODALITIES

A key reason for recent interest in SAEs is the observation that the learned dictionary elements (or equivalently, the neurons of the latent space) often seem to correspond to interpretable concepts or ideas. Before proceeding to our main results, we first define an evaluation metric which we will use to jointly evaluate the *semanticity* and *multimodality* of each coordinate of the SAE's latent space. Our metric generalizes the main idea behind the monosemanticity score proposed in (Pach et al., 2025) to measure semanticity with respect to a pair of modalities. For each neuron and any pair of modalities $(m, n)$ (e.g., $m = image$ and $n = text$), we define the multimodal monosemanticity score (MMS) with respect to $(m, n)$ as follows:

1. On an unseen validation dataset, find all samples from each modality with non-zero activations for the given neuron. Store these activations as vectors $\boldsymbol{a}^{(m)} \in \mathbb{R}^M$ and $\boldsymbol{a}^{(n)} \in \mathbb{R}^N$.

2. Let $\boldsymbol{S} \in \mathbb{R}^{M \times N}$ be the matrix of cosine similarities of these samples under a separate multimodal encoder (i.e., a different encoder than the one used in learning the dictionary).

3. Compute the co-activation matrix $\boldsymbol{A} \in \mathbb{R}^{M \times N}$, where $A_{ij} = |a_i^{(m)} a_j^{(n)}|(1 - \delta_{mn})$, where $\delta_{mn} = 1$ if $m = n$, else 0.

4. Normalize co-activations to sum to 1: $\tilde{A}_{ij} = \frac{1}{\sum_{i,j} A_{ij}} A_{ij}$.

5. Compute $MMS(m, n)$ as the weighted sum of cosine similarities, with weights given by normalized co-activations:

$$MMS(m, n) = \sum_{i,j} \tilde{A}_{ij} S_{ij}$$

The high-level intuition for this metric is that a neuron should be considered more monosemantic if it is active for inputs that are semantically similar to each other. In addition to capturing the semanticity of each SAE neuron, the case $m \neq n$ captures the multimodality of a concept. In particular, if a dictionary is fully modality-split (cf. Definition 1), then there are no co-activating inputs and $MMS(m, n)$ trivially equals 0. A positive $MMS(m, n)$ scores indicates that there are test samples of different modalities that co-activate for a given neuron and that concept can be considered more multimodal. In Section 5, we will see that our proposed methodologies for multimodal SAEs increase the $MMS$ score significantly compared to classical SAEs.

**Comparison to metrics proposed in the literature:** Our metric is most similar to the MS score in (Pach et al., 2025), but it also allows monosemanticity to be measured between two different modalities. Additionally, we normalize the co-activations to sum to 1, rather than using min-max normalization on the activations themselves. This allows us to interpret the metric more naturally as a weighted average of cosine similarities. We also note that our MMS metric measures a fundamentally different quantity than the BridgeScore in (Papadimitriou et al., 2025). The BridgeScore assigns a value to each pair of neurons/concepts (answering the question of how often neuron $i$ and neuron $j$ are jointly active across paired samples), while our MMS score assigns a value to a single neuron/concept (answering the question of how often a single neuron co-activates for semantically similar inputs of different modalities). Additionally, we measure semanticity by using cosine similarity as a proxy, which obviates the need for additional paired data to compute the metric.

## 4 LEARNING MULTIMODAL CONCEPTS WITH GROUP-SPARSE AUTOENCODERS

In this section, we first argue that a modality-split dictionary on an aligned embedding space can be improved to a multimodal dictionary with similar reconstruction loss and strictly better multimodal alignment.

This result is detailed below in Theorem 1, and we defer the proof to Appendix A.1.

**Theorem 1.** *Consider a set of n paired unit-vector embeddings corresponding to data from different modalities $\{(x^{(i)}, y^{(i)})\}_{i=1}^n$. Suppose the following conditions are met:*

1. ***Paired embeddings are aligned:*** *There is a $c > 0$ such that $\langle x^{(i)}, y^{(i)} \rangle > c$ for all $i \in [n]$.*

2. ***Sparse decomposition in a split dictionary:*** *There exists a modality-split dictionary $\boldsymbol{W} \in \mathbb{R}^{d \times p}$ satisfying the following: for all $\boldsymbol{v} \in \{x_1, y_1, \ldots, x_n, y_n\}$ there is a $K$-sparse vector $\boldsymbol{z}_v$ with non-negative entries such that $\boldsymbol{W}\boldsymbol{z}_v = \boldsymbol{v}$.*

*Then, there exists a dictionary $\tilde{\boldsymbol{W}}$ of size $p + n$ admitting a $(K + 1)$-sparse decomposition of all $2n$ embeddings, and where sparse codes of all pairs have strictly positive inner product.*

Here, the first assumption follows naturally from the fact that we are considering dictionaries learned from aligned embedding spaces, where embeddings of paired samples containing similar semantic information should be positively correlated. We also note that the second condition can be relaxed in a straightforward way to the case where sparse decompositions under $\boldsymbol{W}$ hold only approximately, up to some small error $\epsilon$, adding $O(\epsilon)$ error terms to the error of the decompositions with respect to $\tilde{\boldsymbol{W}}$. The theorem states that a modality-split dictionary can always be improved to a dictionary where sparse codes are more aligned between modalities. Given this result, we turn to the question of how to induce a more favorable implicit bias in SAEs training on multimodal embeddings.

To this end, we propose a new training paradigm that can be applied to a training set of paired multimodal samples. We first define the *group-sparse* loss that we will use to learn multimodal dictionaries. The group-sparse loss corresponding to two sparse codes $\boldsymbol{z}$ and $\boldsymbol{w}$ is given by

$$\mathcal{L}_{gs}(\boldsymbol{z}, \boldsymbol{w}) = \left\| \begin{bmatrix} \boldsymbol{z}^\top \\ \boldsymbol{w}^\top \end{bmatrix} \right\|_{2,1} = \sum_{i=1}^{p} \sqrt{z_i^2 + w_i^2}.$$

The use of the convex $L_{2,1}$ norm to enforce structured sparsity in parameters has been studied extensively in the context of problems like the group LASSO (Yuan & Lin, 2006) and multi-task learning (Argyriou et al., 2008). Intuitively, this loss function encourages coordinates of $\boldsymbol{z}$ and $\boldsymbol{w}$ to be *jointly sparse*, i.e., have the same support. Although we focus here on the case of two modalities, we note that the overall formulation naturally extends to an arbitrary number of modalities.

Given paired embeddings $\boldsymbol{x}, \boldsymbol{y} \in \mathbb{R}^d$ (corresponding to semantically similar data from different modalities), our model takes the form:

$$\begin{aligned} \boldsymbol{z}_x &= \text{TopK}(\text{ReLU}(\boldsymbol{W}_{enc}(\boldsymbol{x} - \boldsymbol{b}_0) + \boldsymbol{b})), & \boldsymbol{z}_y &= \text{TopK}(\text{ReLU}(\boldsymbol{W}_{enc}(\boldsymbol{y} - \boldsymbol{b}_1) + \boldsymbol{b})), \\ \hat{\boldsymbol{x}} &= \boldsymbol{W}_{dec}\boldsymbol{z}_x + \boldsymbol{b}_0, & \hat{\boldsymbol{y}} &= \boldsymbol{W}_{dec}\boldsymbol{z}_y + \boldsymbol{b}_1. \end{aligned}$$

Here, $\boldsymbol{b}_0$ and $\boldsymbol{b}_1$ are learnable pre-coding bias terms and the remaining parameters are shared across modalities. Similar to the standard SAE formulation, our approach trains an SAE to reconstruct embeddings from different data modalities, but we also add the group-sparse loss term specified above with a regularization hyperparameter $\lambda > 0$, to encourage $\boldsymbol{z}_x$ and $\boldsymbol{z}_y$ to have shared sparsity structure:

$$\mathcal{L} = \|\boldsymbol{x} - \hat{\boldsymbol{x}}\|_2^2 + \|\boldsymbol{y} - \hat{\boldsymbol{y}}\|_2^2 + \lambda \mathcal{L}_{gs}(\boldsymbol{z}_x, \boldsymbol{z}_y).$$

Additionally, to mitigate the occurrence of dead neurons (dictionary elements which never activate for any inputs) and to further encourage multimodality of concepts, we also introduce *cross-modal random masking* during training. Specifically, we apply a shared random mask with probability $p$ during the computation of $\boldsymbol{z}_x$ and $\boldsymbol{z}_y$ prior to the TopK operation. This forces the TopK operation to choose from the same subset of coordinates at each step prior to decoding. The full training approach is summarized in Figure 2.

## 5 RESULTS

To evaluate our approach, in this section we compare three main types of SAE training schemes:

- **SAE:** Standard TopK SAE (Gao et al., 2024) trained on reconstruction loss of multimodal embeddings.
- **GSAE:** Group-sparse autoencoder trained on both reconstruction and group-sparsity losses, but without random masking.
- **MGSAE:** Masked, group-sparse autoencoder trained using our complete training pipeline from Figure 2.

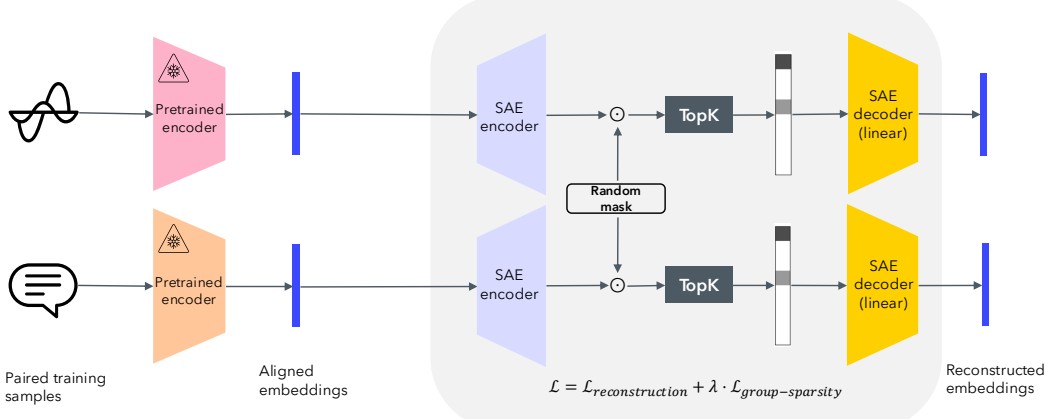

Figure 2: **Masked group-sparse autoencoder for multimodal concept extraction:** Our proposed approach is trained on paired data. Embeddings are encoded to a higher-dimensional vector, masked (with the same mask for each modality), sparsified using TopK, and decoded with a linear layer.

We train each of these variants on two different multimodal embedding spaces: (1) CLIP ViT-B/16 (Radford et al., 2021) embeddings obtained from the CC3M dataset of image/text caption pairs (Sharma et al., 2018) and (2) LAION CLAP embeddings (Wu et al., 2023) obtained from the JamendoMaxCaps dataset of music/text pairs (Roy et al., 2025). In the latter setting, we use a CLAP checkpoint specifically finetuned on music data. To our knowledge, ours is the first work to apply SAEs to the joint embedding space of music/text samples and measure semanticity of the learned dictionary. In each case we fix the sparsity level $K = 32$ and the dictionary size as $p = 16d$, where $d = 512$ is the dimension of the original CLIP/CLAP embedding. We train for 25,000 steps in the image/text setting and 10,000 steps in the music/text setting (chosen to ensure convergence in the fraction of explained variance for both modalities). All embeddings are normalized to have unit norm prior to training. For further experimental details, see Appendix A.2. In Section 5.2, we also provide additional comparisons to the BatchTopK (Bussmann et al., 2024) and Matryoshka (Zaigrajew et al., 2025) SAE variants, which have both been applied to CLIP embeddings in previous works (Papadimitriou et al., 2025; Pach et al., 2025). While we focus on CLIP and CLAP embeddings in the main paper for clarity of exposition, we also provide additional experiments on the SIGLIP2 (Tschannen et al., 2025)and AIMv2 (Fini et al., 2024) text/image encoders in Appendix A.3 to demonstrate that our methods generalize to other popular multimodal embedding spaces.

## 5.1 DEAD NEURONS AND MONOSEMANTICITY

We first evaluate our approach by considering the prevalence of dead neurons in a *modality-specific* way. In other words, given an unseen validation set of paired samples, we compute how many neurons are active for at least one sample from a given modality. For the image/text case, we use 10,000 pairs from the validation set of CC3M, and for the music/text case, we use 10,000 pairs from the MusicBench

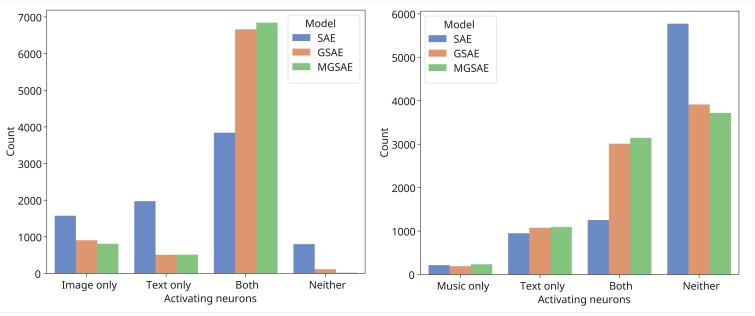

Figure 3: Number of neurons activating for each individual modality, both modalities, and neither modality. Left: models trained on CLIP embeddings, validation on CC3M val. set. Right: models trained on CLAP embeddings, validation on MusicBench.

dataset (Melechovsky et al., 2024). We visualize the number of dead neurons in Figure 3. We can

see that, compared to the standard (TopK) SAE, the GSAE and MGSAE both have a significant increase in the number of neurons that activate for both modalities and a reduction in the number of neurons that activate for neither modality. Overall, the MGSAE consistently obtains the highest number of multimodal activations and smallest number of dead neurons.

On the same validation sets, we also compute the multimodal monosemanticity score we defined in Section 3.2 between all pairs of modalities: $MMS(image, image)$, $MMS(text, text)$, $MMS(image, text)$ (and likewise for audio/text). We recall here that a high MMS score for a given neuron indicates that inputs which activate that neuron are similar to each other (and hence that neuron encodes a coherent concept). To compute the similarity matrix between validation samples, we use the CLIP RN-50 encoder for CC3M and Microsoft CLAP (Elizalde et al., 2023) for MusicBench. Our results, plotted in Figure 4, confirm the notion that standard SAEs have a large proportion of neurons that do not activate across different modalities (and a large number of dead neurons more generally). By contrast, we see that the GSAE and, to an even greater extent, the MGSAE, both have a much larger proportion of neurons with high monosemanticity scores relative to the baseline score achieved by the original dense CLIP/CLAP embeddings. Our results indicate that the group-sparse variants are *more semantic and more multimodal* than the standard SAE in these settings.

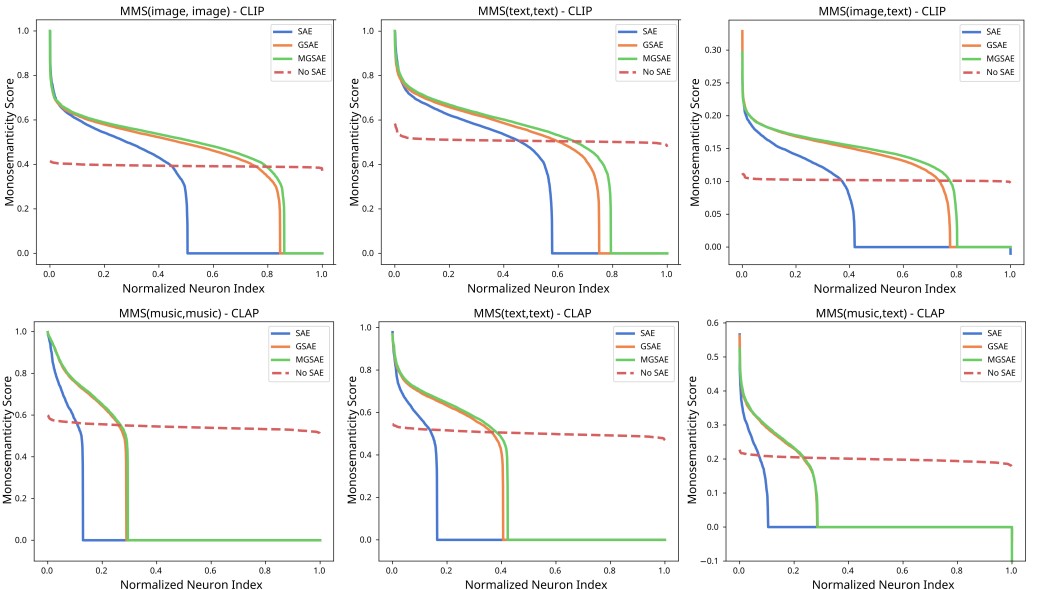

Figure 4: Multimodal monosemanticity (MMS) scores for each feature, arranged in descending order (higher is better). Neuron index (on the horizontal axis) is normalized due the difference in dimensionality between the "No SAE" case ($d = 512$) and the SAE variants ($d = 16 \cdot 512$). Top row: MMS scores for models trained on CLIP embeddings for image/text data. Bottom row: MMS scores for models trained on CLAP embeddings for music/text data.

## 5.2 ZERO-SHOT PERFORMANCE ON CROSS-MODAL TASKS

Another way to evaluate the ability of SAEs to learn multimodal dictionaries is to measure zero-shot performance of sparse codes on cross-modal tasks. We note that the purpose of these evaluations is *not* to claim that sparse codes themselves constitute a good embedding which should be used zero-shot tasks; instead, we use these metrics as a proxy to measure how much modality alignment is preserved when extracting sparse features using SAE-based models. Indeed, for any dictionary with a small number of multimodal concepts, the sparse codes corresponding to paired embeddings from different modalities have low cosine similarity, so we expect zero-shot tasks to perform quite poorly (in the extreme case of a modality-split dictionary, the cosine similarity is always zero and zero-shot tasks are impossible). By contrast, if embeddings are decomposed into concepts which depend on semantic information in a modality-agnostic way, then we would expect zero-shot performance to be retained to a larger extent.

Table 1: Performance comparison on zero-shot image/text cross-modal tasks between SAE (3 types), GSAE, and MGSAE. Original performance on the dense CLIP embeddings is provided for reference. Reported numbers are classification accuracy. Numbers with an asterisk indicate results reported from the original authors of the model.

| Model | ZS Image/Text Tasks | | |
|---|---|---|---|
| | CIFAR-10 | CIFAR-100 | ImageNet |
| SAE - TopK (Gao et al., 2024) | 0.657 | 0.418 | 0.303 |
| BatchTopK SAE (Bussmann et al., 2024) | 0.657 | 0.277 | 0.178 |
| Matryoshka SAE (Zaigrajew et al., 2025) | 0.587 | 0.166 | 0.185 |
| GSAE (ours) | 0.808 | 0.526 | 0.354 |
| MGSAE (ours) | **0.842** | **0.554** | **0.373** |
| *CLIP ViT B/16* | 0.916* | 0.687* | 0.686* |

Table 2: Performance comparison on zero-shot audio/text cross-modal tasks between SAE, GSAE, and MGSAE. Original performance on the dense CLAP embeddings is provided for reference. Reported numbers are classification accuracy or mean reciprocal rank (for FMACaps). Numbers with an asterisk indicate results reported from the original authors of the model.

| Model | ZS Audio/Text Tasks | | |
|---|---|---|---|
| | GTZAN Genres | NSynth Instruments | FMACaps retrieval |
| SAE - TopK (Gao et al., 2024) | 0.376 | 0.265 | 0.023 |
| GSAE (ours) | **0.705** | 0.303 | 0.050 |
| MGSAE (ours) | 0.672 | **0.354** | **0.061** |
| *LAION CLAP* | 0.710* | 0.339 | 0.075 |

In Tables 1 and 2, we state results for zero-shot performance on a variety of tasks for each multimodal setting. For the models trained on the CLIP embedding space, we consider zero-shot classification on the CIFAR-10 (Krizhevsky et al., 2009), CIFAR-100 (Krizhevsky et al., 2009), and ImageNet (Deng et al., 2009) datasets. In all cases, the standard TopK SAE (together with recently proposed variants like the BatchTopK and Matryoshka SAE) has a drastic performance reduction on zero-shot tasks compared to the original CLIP embeddings, while the group-sparse variants demonstrate a marked improvement over the SAE of almost 20 percent for CIFAR-10, 15 percent for CIFAR-100, and 7 percent for ImageNet.

For models trained on the CLAP embedding space, we consider zero-shot classification on the GTZAN genres (Tzanetakis & Cook, 2002) and NSynth instruments (Engel et al., 2017) datasets and zero-shot text-to-music retrieval on the FMACaps dataset (Melechovsky et al., 2024) (with performance measured as mean reciprocal rank). For these tasks, the group-sparse variants also significantly outperform standard SAEs, in some cases performing close to twice as well. Interestingly, we find that in the music/text setting, the sparse codes are actually quite competitive with (and on one task, even better than) the original dense CLAP embeddings, despite being 16 times more sparse and more semantic, as measured by the metrics in the previous section.

## 5.3 CASE STUDY: CONCEPT NAMING AND INTERPRETING LINEAR PROBES

In the previous sections, we showed that group-sparse autoencoders can improve multimodal alignment and semanticity in the dictionaries learned by sparse autoencoders. Next, we show a simple application to illustrate the limitations of classical SAEs for interpretability studies in multimodal spaces and the advantages of the group-sparsity approach we propose. We consider binary classification on the CelebA dataset (Liu et al., 2015). We first train a simple linear classifier on the CLIP ViT-B/16 embeddings of the training set to predict the presence of blonde hair. Given this trained linear probe, denoted $\hat{\theta}$, a natural application of SAEs is to identify which semantic properties of a photo contribute most to it being classified as "blonde" (and to detect the presence of potential

spurious features). To do this, we use the following procedure, where $W$ is the learned dictionary matrix (with normalized columns):

1. **Concept naming:** We follow the approach used in multiple recent works for interpreting CLIP embeddings (Rao et al., 2024; Zaigrajew et al., 2025). In words, we use a large vocabulary corpus $\mathcal{V}$ of potential concept names and assign each dictionary vector the word or phrase in $\mathcal{V}$ with highest CLIP cosine similarity (See Appendix A.2 for more details).

2. **Compute $\hat{\theta}$'s coefficients in the concept dictionary:** The entries of $\alpha = W^{\top}\hat{\theta}$ give the contribution of each concept to the prediction of the classifier. The concept names corresponding to the largest entries of $\alpha$ can be interpreted as the concepts with highest contribution to a positive classification (in our example, "blonde").

As shown in Figure 5, the above procedure applied with the MGSAE dictionary yields dominating concepts which are highly correlated with the true label being predicted ("blonde") and offers a suggestion of spurious correlations with gender ("girl", "woman"). This is consistent with known results on that gender is a spurious correlation for the blonde hair attribute in CelebA (Liu et al., 2015). The SAE trained without group-sparsity, by contrast, is much less successful at identifying which concepts are important for this task, which we attribute to the fact that the success of the concept naming step relies crucially on having a multimodal dictionary. In Appendix A.5 we provide an additional example of how multimodal dictionaries can be useful for the task of steering retrieval systems.

Figure 5: Leading concepts contributing to a classification of "blonde" on the CelebA dataset, extracted using a standard SAE (left) and MGSAE (right).

## 6 CONCLUSION

Based on the observation that standard SAEs tend to learn many unimodal dictionary elements, we propose a new method for learning sparse dictionary decompositions of multimodal embedding spaces. Our method crucially makes a connection between *multimodality* of the dictionary elements and *group-sparsity* of the sparse codes corresponding to paired data of different modalities. We encourage this group-sparse property during training both through explicit regularization (inspired by classical techniques in sparse recovery) and cross-modal random masking. We show through a variety of metrics that our results learn more semantic and aligned embeddings than standard SAEs, and we demonstrate that our method can be more effective for interpretability analysis and identifying spurious correlations, since improved modality alignment leads to more accurate concept naming. Our results promote improved interpretability and control of multimodal representation spaces and lay the groundwork for further research on improving alignment in multimodal SAEs. In particular, we note that the group-sparse loss and the masking technique we employ have straightforward extensions to settings with more than two modalities. Additionally, our proposed methodology can also be adapted to settings where large amounts of unpaired data are also available. In such settings, the group-sparse loss can be applied only for paired samples, and reconstruction loss alone can be used for the remaining data samples. While we leave a detailed investigation of this for future work, we believe that following this approach with even a small amount of paired data would provide improvements over the baseline SAE.

### ACKNOWLEDGMENTS

Support for this project was generously provided by Dolby Laboratories. We also thank Jacob Abernethy for compute assistance.

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

# A APPENDIX

## A.1 PROOF OF THEOREM 1

*Proof.* We can assume that the columns of $W$ are unit-norm, by appropriately scaling the sparse codes of each embedding in the set. Take any pair of embeddings; for simplicity of notation, call these embeddings $x$ and $y$. By the two conditions in the theorem statement, we know that $\langle x, y \rangle > c$ and there are $K$ sparse vectors $z_x$ and $z_y$ with non-negative entries such that $x = W z_x$ and $y = W z_y$. Since $W$ is modality-split, $z_x$ and $z_y$ have disjoint support, so their inner product is 0 and

$$\mathcal{L}_{gs}(z_x, z_y) = \|z_x\|_1 + \|z_y\|_1.$$

Without loss of generality, we can assume the support of these two sparse vectors is $\{1, \ldots, K\}$ and $\{K+1, \ldots, 2K\}$. Then, we have

$$c < \langle \boldsymbol{z}_x, \boldsymbol{x}_y \rangle = \sum_{i=1}^{K} \sum_{j=K+1}^{2K} z_{x,i} z_{y,j} \boldsymbol{w}_i^\top \boldsymbol{w}_j.$$

Assume for the sake of contradiction that $\boldsymbol{w}_i^\top \boldsymbol{w}_j < \frac{c}{\|\boldsymbol{z}_x\|_1 \|\boldsymbol{z}_y\|_1}$ for all $(i,j)$ pairs in the double summation above. Then, we would have

$$\sum_{i=1}^{K} \sum_{j=K+1}^{2K} z_{x,i} z_{y,j} \boldsymbol{w}_i^\top \boldsymbol{w}_j < \sum_{i=1}^{K} \sum_{j=K+1}^{2K} z_{x,i} z_{y,j} \frac{c}{\|\boldsymbol{z}_x\|_1 \|\boldsymbol{z}_y\|_1} = c,$$

which is a contradiction. Hence, there exists some $(i,j)$ with $i \in \{1, \ldots, K\}$ and $j \in \{K+1, \ldots, 2K\}$ satisfying $\boldsymbol{w}_i^\top \boldsymbol{w}_j \geq \frac{c}{\|\boldsymbol{z}_x\|_1 \|\boldsymbol{z}_y\|_1}$. We can assume WLOG that these are $\boldsymbol{w}_1$ and $\boldsymbol{w}_{K+1}$.

The construction of $\tilde{\boldsymbol{W}}$ then follows naturally from the Gram-Schmidt orthogonalization process. Define the matrix $\tilde{\boldsymbol{W}} = [\boldsymbol{w}_1, \ldots, \boldsymbol{w}_K, \boldsymbol{w}_{K+1}, \boldsymbol{w}_{K+2}, \ldots, \boldsymbol{w}_p, \tilde{\boldsymbol{w}}_{K+1}]$, where

$$\tilde{\boldsymbol{w}}_{K+1} := \frac{\boldsymbol{w}_{K+1} - (\boldsymbol{w}_1^\top \boldsymbol{w}_{K+1}) \boldsymbol{w}_1}{\|\boldsymbol{w}_{K+1} - (\boldsymbol{w}_1^\top \boldsymbol{w}_{K+1}) \boldsymbol{w}_1\|_2}$$

In this new dictionary, all embeddings besides $\boldsymbol{y}$ admit the same sparse code with a $0$ appended in the $p+1$th position. A decomposition of $\boldsymbol{y}$ in the dictionary $\tilde{\boldsymbol{W}}$ can be computed as

$$\boldsymbol{y} = z_{y,K+1} \boldsymbol{w}_{K+1} + \sum_{j=K+2}^{2K} z_{y,j} \boldsymbol{w}_j$$

$$= z_{y,K+1} \big( \tilde{\boldsymbol{w}}_{K+1} \|\boldsymbol{w}_{K+1} - (\boldsymbol{w}_1^\top \boldsymbol{w}_{K+1}) \boldsymbol{w}_1\|_2 + (\boldsymbol{w}_1^\top \boldsymbol{w}_{K+1}) \boldsymbol{w}_1 \big) + \sum_{j=K+2}^{2K} z_{y,j} \boldsymbol{w}_j$$

$$= z_{y,K+1} \boldsymbol{w}_1^\top \boldsymbol{w}_{K+1} \boldsymbol{w}_1 + z_{y,K+1} \|\boldsymbol{w}_{K+1} - (\boldsymbol{w}_1^\top \boldsymbol{w}_{K+1}) \boldsymbol{w}_1\|_2 \tilde{\boldsymbol{w}}_{K+1} + \sum_{j=K+2}^{2K} z_{y,j} \boldsymbol{w}_j.$$

Letting the sparse codes of $\boldsymbol{x}$ and $\boldsymbol{y}$ under $\tilde{\boldsymbol{W}}$ be denotes as $\tilde{\boldsymbol{z}}_x$ and $\tilde{\boldsymbol{z}}_y$, respectively, we can therefore conclude

$$\langle \tilde{\boldsymbol{z}}_x, \tilde{\boldsymbol{z}}_y \rangle = \langle \boldsymbol{z}_x, \tilde{\boldsymbol{z}}_y \rangle = z_{x,1} z_{y,K+1} \boldsymbol{w}_1^\top \boldsymbol{w}_{K+1} \geq \frac{z_{x,1} z_{y,K+1}}{\|\boldsymbol{z}_x\|_1 \|\boldsymbol{z}_y\|_1} \cdot c > 0.$$

Noting that we can apply the same argument iteratively to increase the alignment of the other pairs by adding a single dictionary element at each step completes the proof.

$\square$

## A.2 EXPERIMENTAL DETAILS

In this section, we provide additional experimental details for the results contained in Section 5.

### A.2.1 SAE TRAINING DETAILS

In the main paper, we train and evaluate each SAE variant in two settings, listed below:

**CLIP ViT-B/16:** We train SAEs on the CLIP embeddings of the CC3M (Sharma et al., 2018) dataset of image/text captions, with a balanced number of samples per modality. We fix the sparsity level $K = 32$ and the expansion factor as $16$, resulting in a dictionary dimensionality of $8192$. We train for $25000$ iterations with a batch size of $128$. We adapt the TopK SAE implementation from (Marks et al., 2024) for our setup, with additional group-sparse regularization and cross-modal random masking. We sweep the group-sparsity regularization parameter over $\lambda \in \{0.01, 0.05, 0.1, 0.2\}$ and choose the largest value ($0.05$) that does not push the average sparsity within a batch below

$K = 32$. We then choose a random mask probability from $\{0.1, 0.2, 0.3, 0.4\}$ such that the fraction of explained variance is similar to the GSAE with $\lambda = 0.05$, and we decide on $p = 0.2$. The learning rate in each case is chosen based on the scaling law in Figure 3 of (Gao et al., 2024) (which is the default learning rate in the library (Marks et al., 2024)), and all models are trained with the Adam optimizer. We train the BatchTopK SAE with the same expansion factor of 16 and a choice of $K = 32 * 128$ for the sparsity level of each batch. For the Matryoshka SAE, we use the checkpoint provided by the authors of (Zaigrajew et al., 2025) with an expansion factor 16, uniform weighting, and inference time value of $K = 32$.

**LAION CLAP:** Parameters are chosen in the same way as in the CLIP case, except we train for only 10000 iterations, noting that the loss converges well before this point. Using the same heuristics for hyperparameter selection, we choose a group-sparsity parameter $\lambda = 0.05$ and random mask parameter $p = 0.1$. The specific checkpoint of CLAP which we use ("music_audioset_epoch_15_esc_90.14.pt") is finetuned for music data, and we train on the Jamendo-MaxCaps dataset (Roy et al., 2025), which we preprocess into pairs of 30 second music clips and corresponding captions (using the provided annotations file).

### A.2.2 EFFECT OF EXPANSION FACTOR AND $K$

In our main experiments, we fix the expansion factor of all SAE variants as 16 and the sparsity level as $K = 32$. In this section, we provide additional experimental results to demonstrate that the key insights in our paper are stable for a wide range of these hyperparameters. Concretely, we sweep different choices of expansion factor and $K$ and train 3 models—SAE, GSAE, and MGSAE—for each configuration, with group-sparsity parameter $\lambda = 0.01$ and mask parameter $p = 0.05$. In Table 3, we report the zero-shot performance on ImageNet for the sparse latents from each of these models. As seen in the table, the MGSAE consistently achieves the highest zero-shot performance, and both variants with the group-sparse loss outperform the baseline SAE in all cases. Interestingly, we find that the expansion factor has minimal effect on multimodal alignment (as measured by zero-shot performance), while $K$ has a significant effect, with larger values leading to better performance.

Table 3: Zero-shot performance on ImageNet for different expansion factor and sparsity level configurations.

| Expansion factor | K | SAE | GSAE | MGSAE |
|---|---|---|---|---|
| 4 | 16 | 0.287 | 0.294 | **0.309** |
| 4 | 32 | 0.335 | 0.361 | **0.381** |
| 4 | 64 | 0.357 | 0.407 | **0.435** |
| 8 | 16 | 0.295 | 0.308 | **0.321** |
| 8 | 32 | 0.322 | 0.349 | **0.371** |
| 8 | 64 | 0.326 | 0.400 | **0.436** |
| 16 | 16 | 0.276 | 0.313 | **0.314** |
| 16 | 32 | 0.295 | 0.341 | **0.361** |
| 16 | 64 | 0.304 | 0.375 | **0.399** |

### A.2.3 LINEAR PROBE ON CELEBA

In this section, we provide additional details on how we use concept dictionaries to interpret the behavior of a linear classifier on the CelebA (Liu et al., 2015) dataset. Suppose $\boldsymbol{W}$ is a learned dictionary which (approximately) admits a sparse decomposition $\boldsymbol{x} \approx \boldsymbol{W}\boldsymbol{z}$ for any embedding $\boldsymbol{x} \in \mathcal{X}$. And $\hat{\boldsymbol{\theta}}$ is the parameter of a trained linear classifier (we use logistic regression with no bias parameter and 2000 iterations of LBFGS).

We first assign a concept name to each column of $\boldsymbol{W}$ using the "Discover-then-Name" approach from (Rao et al., 2024). In particular, let $\mathcal{V}$ be a large vocabulary of words. Then, for each column $\boldsymbol{w}_i$ of $\boldsymbol{W}$, we assign a concept name using:

$$\text{Concept}(i) = \arg\max_{v \in \mathcal{V}} \left\langle \frac{\boldsymbol{w}_i}{\|\boldsymbol{w}_i\|_2}, f(v) \right\rangle,$$

where $f$ is the CLIP text encoder. In words, each dictionary element is assigned the label of the word in the vocabulary with the most similar CLIP embedding. We use a vocabulary provided in the implementation of (Bhalla et al., 2024a), where $\mathcal{V}$ is the set of 10000 most frequent tokens in the LAION-400m dataset and 5000 of the most frequent bigrams in the LAION-400m dataset.

Given these concept labels for each $\boldsymbol{w}_i$, we proceed to the analysis of the linear probe $\hat{\boldsymbol{\theta}}$. In particular, note that a positive label ("blonde") is given to test samples $\boldsymbol{x}$ for which $\hat{\boldsymbol{\theta}}^\top \boldsymbol{x}$ is large. Decomposing this in the dictionary, we obtain

$$\hat{\boldsymbol{\theta}}^\top \boldsymbol{x} \approx \sum_{i=1}^{p} z_i \boldsymbol{w}_i^\top \hat{\boldsymbol{\theta}}.$$

Hence we see that when $\boldsymbol{w}_i^\top \hat{\boldsymbol{\theta}}$ is large, concept $i$ is weighted more heavily by the classifier. The concepts listed in Figure 5 are the those corresponding to largest seven values of $\boldsymbol{w}_i^\top \hat{\boldsymbol{\theta}}$

### A.3 FURTHER EXPERIMENTS ON SIGLIP2 AND AIMV2

To show that our proposed methodologies yield benefits across a wide range of multimodal settings, we also report experimental results for SAE, GSAE, and MGSAE models trained on the SIGLIP2 (Tschannen et al., 2025) and AIMv2 (Fini et al., 2024) text/image encoders. All GSAE models are trained using a group-sparsity parameter of 0.01 and all MGSAE models are trained with a mask parameter of 0.05. The expansion ratio and choice of $k$ are fixed as in our CLIP experiments. For each encoder and each SAE variant, we report the dead neuron counts for each modality and the performance on three cross-modal tasks. We see that across both encoders, the MGSAE achieves the highest number of multimodal neurons and the smallest number of dead neurons. In the zero-shot tasks, the GSAE and MGSAE both consistently outperform the SAE while having similar performance to each other. As in the main paper, the dead neuron counts are computed using the validation set of CC3M.

Table 4: Dead neuron counts for models trained on SIGLIP2 ViT-B/16 embeddings of the CC3M dataset.

| Model | Image only | Text only | Both | Neither |
|-------|-----------|-----------|------|---------|
| SAE | 1590 | 2234 | 3206 | 5258 |
| GSAE | 1286 | 2140 | 6584 | 2278 |
| MGSAE | 1346 | 2095 | **6972** | **1875** |

Table 5: Performance comparison on zero-shot cross-modal tasks between SAE, GSAE, and MGSAE, all trained on SIGLIP2 embeddings. Original performance on the dense SIGLIP2 embeddings is provided for reference.

| Model | ZS Image/Text Tasks | | |
|-------|----------|-----------|----------|
| | CIFAR-10 | CIFAR-100 | ImageNet |
| SAE | 0.918 | 0.608 | 0.451 |
| GSAE | **0.922** | **0.629** | 0.477 |
| MGSAE | 0.906 | 0.628 | **0.481** |
| *SIGLIP2 ViT B/16* | 0.939 | 0.739 | 0.686 |

Table 6: Dead neuron counts for models trained on AIMv2 embeddings of the CC3M dataset.

| Model | Image only | Text only | Both | Neither |
|-------|-----------|-----------|------|---------|
| SAE | 232 | 231 | 2152 | 9673 |
| GSAE | 185 | 192 | 3572 | 8339 |
| MGSAE | 185 | 216 | **3910** | **7977** |

Table 7: Performance comparison on zero-shot cross-modal tasks between SAE, GSAE, and MGSAE, all trained on AIMv2 embeddings. Original performance on the dense AIMv2 embeddings is provided for reference.

| Model | ZS Image/Text Tasks | | |
|---|---|---|---|
| | CIFAR-10 | CIFAR-100 | ImageNet |
| SAE | 0.963 | 0.739 | 0.443 |
| GSAE | **0.972** | 0.799 | 0.493 |
| MGSAE | 0.970 | **0.802** | **0.522** |
| *AIMv2* | 0.976 | 0.858 | 0.745 |

## A.4 ADDITIONAL EXPERIMENTAL RESULTS ON MS COCO

To demonstrate the robustness of our approach, we also provide results obtained by training the three model variants we consider in Section 5 on a different dataset, MS COCO (Lin et al., 2014), consisting of 330000 images with captions. In Table 8, we provide neuron counts per modality, to show that the group-sparse variants contain more multimodal concepts than the standard SAE implementation. In Figure 6, we plot the MMS scores for each modality pair. Both of these results are calculated using an unseen collection of 10000 (image,text) pairs from the MS COCO validation set. Since MS COCO contains multiple captions per image, we balance the modalities prior to training to ensure that each model is trained on the same number of image embeddings and text embeddings. These results are consistent with the findings on CC3M, indicating that our models (GSAE and MGSAE) are able to learn a larger number of semantic concepts, particularly in the (text,text) and (image,text) settings.

Table 8: Dead neuron counts for models trained on CLIP ViT-B/16 embeddings of the MS COCO dataset.

| Model | Image only | Text only | Both | Neither |
|---|---|---|---|---|
| SAE | 2118 | 652 | 5364 | 58 |
| GSAE | 625 | 541 | 6987 | 39 |
| MGSAE | 603 | 635 | 6909 | 45 |

Table 9: Zero-shot performance of sparse codes for models trained on CLIP ViT-B/16 embeddings of the MS COCO dataset.

| Model | CIFAR-10 | CIFAR-100 | ImageNet |
|---|---|---|---|
| SAE | 0.547 | 0.341 | 0.189 |
| GSAE | 0.805 | 0.403 | 0.237 |
| MGSAE | **0.820** | **0.423** | **0.247** |

Next, we also report the zero-shot performance of sparse codes for for these models, listed in Table 9. As in the case of models trained on CC3M, we find that the MGSAE obtains the best performance overall, and the group-sparse variants both significantly outperform standard SAEs.

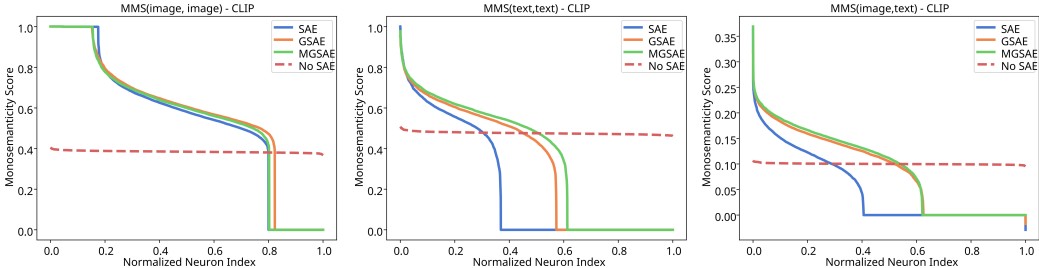

Figure 6: Multimodal monosemanticity score (MMS) for models trained on the MS COCO training dataset. Validation is performed using the MS COCO validation set.

## A.5 STEERING EXAMPLE ON CLAP EMBEDDINGS

In addition to our main contribution of proposing improved dictionary learning techniques for multimodal embedding spaces, our work is to our knowledge the first to apply and interpret SAEs in the context of aligned audio/text embedding spaces like CLAP. So, in this section, we provide a simple example demonstrating the utility of multimodal dictionary decompositions in this domain. We consider a zero-shot text-to-music retrieval problem on the MusicBench dataset (Melechovsky et al., 2024), where a text input is matched to the most similar CLAP embedding of all music files in the dataset. We demonstrate in Figure 7 how interventions using the MGSAE can be used to steer the behavior of this task. We steer using the following process:

1. **Concept naming:** We follow the method of (Rao et al., 2024) to assign a concept name to each dictionary element. For this, we use the Google 20k word corpus as the concept vocabulary and LAION-CLAP similarities to assign concepts to dictionary elements. Using this method, we identify a neuron labeled with the concept "violin".

2. **Intervene on sparse codes of the input prompt:** We take the input prompt "A peaceful song", encode it to a sparse code using the MGSAE, intervene on the "violin" neuron by adding values of 0.5 (Low) and 1.5 (High), decode to a dense vector, and then perform zero-shot retrieval on MusicBench.

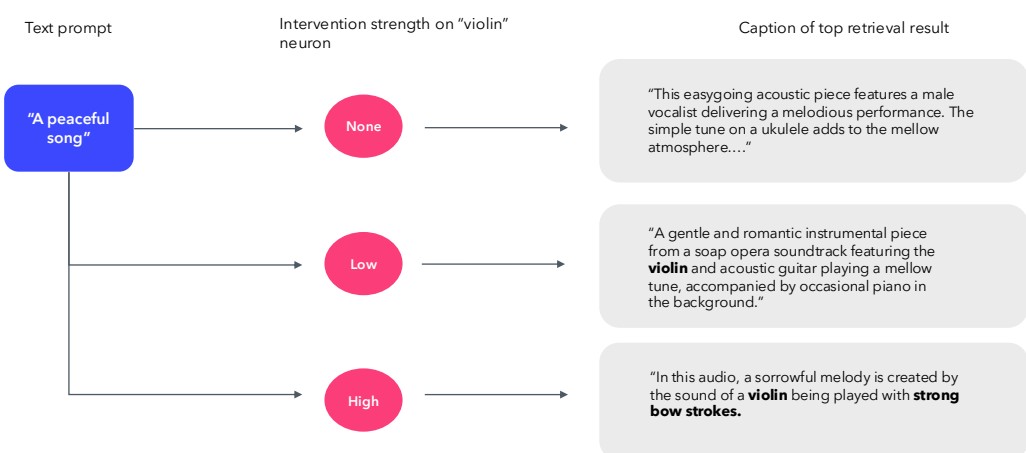

Figure 7: Top retrieval result from MusicBench for the input prompt "A peaceful song" using zero-shot retrieval on CLAP embeddings. We vary the intervention strength on the "violin" neuron and report the caption of the top retrieved music clip.

As shown in the figure, increasing the intervention strength leads to a corresponding emphasis on violin in the top retrieval result, from no violin (in the original retrieval) to a recording with violin in the background, to a recording featuring a violin as the main instrument. We are similarly able to identify several neurons corresponding to genre names ("reggae", "jazz") and instrument types ("organ", "fiddle") that effectively steer retrieval results towards certain concepts. In general, we hope that our preliminary results in this multimodal setting can lead to more interpretable and controllable systems for audio/text, as well as further research into the unique requirements and challenges of SAEs in domains aside from vision-language.

