# OpenReview forum: "Learning multimodal dictionary decompositions with group-sparse autoencoders"
_ICLR.cc/2026/Conference — ICLR 2026 Poster_

### Official Review · Reviewer_xuXW · 2025-10-31

**Soundness:** 3
**Presentation:** 3
**Contribution:** 3
**Rating:** 6
**Confidence:** 4

**Summary:**

The paper “Learning Multimodal Dictionary Decompositions with Group-Sparse Autoencoders” introduces a new approach to understanding multimodal embeddings (such as CLIP or CLAP) through sparse autoencoders (SAEs). While traditional SAEs have been effective for interpreting single-modality models by decomposing embeddings into sparse, interpretable features, they struggle in multimodal settings—often producing “split dictionaries” where features activate for only one modality. The authors address this limitation by theoretically showing that a non-split, multimodal dictionary with better modality alignment can always exist, motivating their proposed model: the Masked Group-Sparse Autoencoder (MGSAE). This model incorporates a group-sparsity regularizer and cross-modal random masking to encourage paired activations between modalities during training, thereby promoting shared multimodal features.

**Strengths:**

1. The paper does more than propose an empirical modification—it provides a theoretical result (Theorem 1) demonstrating that a modality-split dictionary can always be improved to a more aligned, multimodal one. This theoretical grounding strengthens the motivation for their method and shows that poor cross-modal alignment in SAEs is not a fundamental limitation, but rather an optimization bias that can be addressed.
2. The introduction of group-sparse regularization combined with cross-modal random masking is elegant and well-motivated. These additions directly tackle the issue of unimodal activation in SAEs, encouraging shared sparse representations across modalities without requiring extra supervision or hand-engineered alignment losses.
3. The paper is the first to systematically analyze and apply sparse autoencoders to audio/text embeddings (CLAP). This extends interpretability research into a new multimodal domain beyond the widely studied CLIP image–text models.
4. The authors introduce new multimodal monosemanticity metrics that quantify neuron-level semantic alignment across modalities. This is an important methodological contribution for future interpretability research, as it offers a way to measure the coherence and cross-modal consistency of learned concepts.
5. The proposed MGSAE not only improves interpretability—by learning more multimodal, semantically coherent features—but also enhances zero-shot cross-modal task performance, showing practical benefits in alignment-sensitive applications.
6. The paper successfully theoretical insights about dictionary decompositions and practical model design choices. The theoretical results directly inform the architecture, and the empirical results validate those theoretical predictions.

**Weaknesses:**

1. Although the paper evaluates both image–text and audio–text embeddings, these experiments are still confined to two-modality settings. The proposed method’s scalability to more complex multimodal spaces (e.g., video–audio–text or vision–language–action models) remains unexplored. Additionally, the experiments are conducted on relatively small or well-curated datasets (CC3M and JamendoMaxCaps), leaving open questions about generalization to noisier or larger-scale data.
2. The training configurations (e.g., 25k–10k steps with fixed dictionary size and sparsity) seem tuned for feasibility rather than large-scale rigor. There is limited discussion of hyperparameter sensitivity (e.g., λ for group sparsity, masking probability, or K-sparsity), which could significantly affect outcomes. Without broader ablations or scalability studies, it’s difficult to assess robustness or reproducibility across architectures and modalities.
3. While Theorem 1 provides an elegant existence result, it is relatively abstract and does not characterize the conditions under which optimization will find such multimodal dictionaries in practice. The theory does not account for the stochastic and non-convex nature of training neural autoencoders, so the link between theory and empirical convergence remains somewhat heuristic.
4. Most evaluations revolve around interpretability metrics (monosemanticity, dead neurons, activation overlap). While important, the work provides limited evidence of improvements in real downstream tasks such as retrieval, captioning, or multimodal reasoning. Demonstrating such task-level benefits would strengthen claims of practical relevance.
5. The group-sparse and shared masking mechanisms force stronger alignment between modalities, which may inadvertently suppress modality-unique features that are still valuable (e.g., color-specific features in vision but not text). The paper does not deeply analyze whether this trade-off affects the diversity or richness of learned representations.
6. While the paper reports quantitative metrics, it provides few qualitative examples of what specific multimodal concepts the MGSAE learns (e.g., visualizing dictionary atoms corresponding to text–image pairs). Such examples would have made the interpretability claims more concrete and convincing.

**Questions:**

1. Your theorem shows that a non-split dictionary with improved modality alignment always exists. Do you have any insights into how this theoretical result translates into practice—for example, what properties of the optimization landscape or initialization help the model actually discover such dictionaries?
2. You base your approach on the Linear Representation Hypothesis. Have you observed any systematic deviations from linearity in multimodal embeddings (e.g., nonlinear interactions between visual and textual features) that limit the effectiveness of linear sparse decompositions?
3. How sensitive is the model’s performance to the choice of the group-sparse regularizer (L₂,₁ norm)? Did you consider other structured penalties, such as mixed ℓ₁/ℓ∞ norms or hierarchical sparsity, to encourage shared but flexible activations?
4. The random masking step seems crucial for reducing dead neurons. Did you explore deterministic or learned masking strategies, or investigate how the masking probability affects multimodal feature sharing?
5. You share encoder and decoder weights across modalities, except for biases. Have you tried partially shared architectures (e.g., modality-specific encoders with shared latent layers)? If so, how does this affect multimodal alignment?
6. How does your method scale with more than two modalities (e.g., image–text–audio)? Would group-sparsity naturally extend to this case, or would you need a modified loss to maintain balanced multimodal alignment?
7. Did you observe high sensitivity to hyperparameters such as λ (for group sparsity), K (for sparsity level), or the masking probability p? Are there guidelines or heuristics you recommend for stable training?
8. Could you share specific examples of learned multimodal concepts (e.g., visualizing dictionary elements that align text like “a dog running” with corresponding visual features)? This would help illustrate what kinds of shared semantics are captured.
9. Does improving multimodal alignment ever come at the cost of losing fine-grained, modality-specific concepts (like visual texture or acoustic timbre)? How do you balance these competing objectives?

---

> ### Author Response · Authors · 2025-11-25
> **Response to Reviewer xuXW (pt. 1)**
>
> Thank you for your detailed comments and positive feedback! We address each of your questions/comments below:
>
> (1) Extensions to more than two modalities:  We agree that this would be a great area for future work, though we limit ourselves to embedding spaces with two modalities in this work (and we have added two additional encoders in the revision). However, the main techniques are straightforward to generalize to the setting with $n$ modalities. The group sparse loss would take $n$ pairedinputs and compute the $L_{2,1}$ of the $p \times n$ dimensional matrix of sparse codes. Similarly, all the $n$ sparse codes can be masked with the same mask during training. We added a remark about this in the revision and leave a full investigation to future work.
>
> (2) Broader range of architectures and hyperparameters: Please see our general response. We have significantly expanded our experimental results to show robustness across new encoders (SIGLIP2 and AIMv2), expansion ratios, and values of $K$. In each case, we note that the GSAE and MGSAE provide marked improvements over standard SAEs. We also added comparisons to two other SAE variants proposed in the literature -- the BatchTopK and Matryoshka SAEs -- and show that our methods have improved modality alignment.
>
> (3) Thanks for the observation! The goal of Theorem 1 is to demonstrate that the emergence of a split-dictionary is not an inherent limitation in multimodal embedding spaces (since any such dictionary implies the existence of a multimodal one that performs better), and to therefore motivate the construction of training schemes that induce a more favorable inductive bias for multimodal dictionaries. We agree that it does not actually prove that such a dictionary is found during training, but we note that such a theory is challenging and not yet well-understood even for standard autoencoders.
>
> (4) It is correct that we focus our evaluations on improved multimodality and interpretability metrics (which are important for their own sake), but we note that, in addition to interpretability, more multimodal SAEs are essential for control/steering of downstream tasks based on concepts. For example, we provide a simple steering case study in the Appendix that would not be possible using a modality-split dictionary. We hope to elaborate on the steering/control aspect in future work, while focusing here on the preliminary problem of learning multimodal dictionaries, which we argue is a necessary first step towards practical applications of SAEs in multimodal settings.
>
> (5) We agree that this is an important point  - modality-specific concepts may indeed be important to retain in many applications. We note that our models encourage multimodal concepts to be learned without forcing this to be the case, and as seen in our dead neuron results, many of the neurons still activate for only one modality. We believe that proper selection of $\lambda$ could potentially provide a good balance between encouraging multimodality while also still allowing for some unimodal concepts if necessary -- we definitely hope to study this more going forward
>
> (6) Thanks for the suggestion! We include the music steering example to give a concrete example of a neuron that encodes semantic concept ("violin") across modalities, but we hope to include an example for the image/text setting in the final revision (for now, we have focused on larger-scale experimental additions as mentioned in the general response)

---

> ### Author Response · Authors · 2025-11-25
> **Response to Reviewer xuXW (pt. 2)**
>
> Response to Questions:
>
> (1) It is difficult to precisely characterize the best way for a model to discover such dictionaries, but we hope to have presented a convincing argument that group-sparse regularization (and masking) provide one helpful inductive bias to learn more aligned dictionaries. Our proposed MMS metric also provides a principled way to measure the ability of a dictionary learning method to learn multimodal and monosemantic dictionaries, and we hope this will be a useful tool for further research.
>
> (2) We agree that the LRH may not exactly capture the way that semantic information is encoded in embeddings (although we have not investigated this in detail here). However, the LRH is a simple ansatz that has proven to be useful in many applications, and we believe that it is nevertheless important and useful (as evidenced by our interpretability and steering studies) to improve existing dictionary learning approaches in multimodal settings.
>
> (3) Great question! At the initial stages of our work, we did actually experiment with hierarchical sparsity regularizers (e.g., sparse in one modality implies sparse in the other), but did not see any noticeable improvements over the simpler group-sparse regularizer. Hence, we decided to focus on L_{2,1} regularizer for our main approach. It would be interesting to investigate the use of hierarchical sparsity regularizers in settings with 3+ modalities -- we hope that our proposed metrics prove useful in evaluations for this.
>
> (4) We did not experiment with learned masking schemes, but that would be a very interesting direction for future work!
>
> (5) Yes! We did initially try using different encoders for each modality, but (somewhat to our surprise) did not see any improvements over shared encoders, so we used shared encoders for our main results. It would be interesting in future work to see if this would lead to better performance if we used much larger training datasets with more data from each modality (perhaps with some combination of paired and unpaired data)
>
> (6) Yes, the group-sparsity loss would extend naturally to the case of $n$ modalities. In particular, let $z^{(1)}, \dots, z^{(n)}$ be the sparse codes corresponding to paired n-tuples of data for each modality. The corresponding group-sparse loss term is $L_{gs} = \sum_{i=1}^p \sqrt{\sum_{j=1}^n z^{(j) 2}_i}$. The masking scheme we propose also would extend in a natural way.
>
> (7) We have added additional experimental results to evaluate the sensitivity to $K$ and the expansion ratio. With regards to $\lambda$ we use the following useful heuristic (described in the Appendix): we monitor the average sparsity across each batch during training and choose $\lambda$ such that the sparsity level does not drop below the chosen sparsity level $K$.
>
> (8) Thanks for the suggestion. We are working on adding this in during the remainder of the rebuttal period! For now, please see the steering example in the Appendix, which demonstrates that a single neuron encodes the concept "violin" across both text and audio modalities.
>
> (9) As mentioned in Part 1 of our response, we believe that appropriate hyperparameter selection for $\lambda$ and $p$ can help with balancing modality specific and multimodal concepts. An intriguing direction for future work could be to study models which allow for a subset of concepts to be unimodal while the rest are encouraged to be shared across modalities. It is a challenging open question to show that any such approaches would quantifiably improve over existing multimodal SAEs (and such evaluation would likely depend very heavily on what the intended application of the SAE is)

---

### Official Review · Reviewer_G3nW · 2025-11-01

**Soundness:** 2
**Presentation:** 3
**Contribution:** 3
**Rating:** 4
**Confidence:** 2

**Summary:**

The authors have tackled the problem of “split dictionary” observed while using SAEs on multimodal models. They propose the existence of a unified dictionary with improved alignment and also a modified architecture of SAE to realise this. This is applied to CLIP and CLAP data. The authors are also the first to use audio as one of the modalities with text. They also propose a paired multimodal monosemanticity metric.

**Strengths:**

1.	The problem is well motivated
2.	The proposed metric calculation is explained well.
3.	The authors have also considered the problem of dead neurons.

**Weaknesses:**

My key concern is that the comparisons are only with SAE and GSAE. Can the authors provide empirical experiments comparing their method against the prior approaches described in the literature [1-4]?

Other questions are detailed in the Questions section.

**Questions:**

1.	The authors propose a loss term with two group sparse codes – are there ablation studies without this loss?
2.	Also is there any way to prevent the $L_{gs}$ term from causing the codes z and w from collapsing to zero vectors? If this happens, the loss term goes back to being like a standard SAE loss.
3.	The authors propose the use of random masking to encourage multimodality of concepts. Are there ablation studies to show this helps?
4.	My key concern is that the comparisons are only with SAE and GSAE. Can the authors provide empirical experiments comparing their method against the prior approaches described in the literature [1-4]?
5.	The authors mention that they are the first to use audio data in a multimodal setting. Please provide insights on why audio data is hard to include in general. How does their approach compare with audio only methods[5]?
6.	In Figure 3, MGSAE does best consistently, is there any intuition for why it is the best?
7.	The approach has been evaluated on CLIP and CLAP, can SigLIP be added? If not, please explain why.

[1] Isabel Papadimitriou, Huangyuan Su, Thomas Fel, Sham Kakade, and Stephanie Gil. Interpreting the linear structure of vision-language model embedding spaces. arXiv preprint arXiv:2504.11695, 2025.

[2] Mateusz Pach, Shyamgopal Karthik, Quentin Bouniot, Serge Belongie, and Zeynep Akata. Sparse autoencoders learn monosemantic features in vision-language models. arXiv preprint arXiv:2504.02821, 2025.

[3] Hanqi Yan, Xiangxiang Cui, Lu Yin, Paul Pu Liang, Yulan He, and Yifei Wang. Multi-faceted multimodal monosemanticity. arXiv preprint arXiv:2502.14888, 2025.

[4] Vladimir Zaigrajew, Hubert Baniecki, and Przemyslaw Biecek. Interpreting CLIP with hierarchical sparse autoencoders. arXiv preprint arXiv:2502.20578, 2025.

[5] Pluth, D., Zhou, Y., & Gurbani, V. K. (2025, February). Sparse Autoencoder Insights on Voice Embeddings. In 2025 Conference on Artificial Intelligence x Multimedia (AIxMM) (pp. 1-6). IEEE.

---

> ### Author Response · Authors · 2025-11-25
> **Response to Reviewer G3nW**
>
> Thank you for your detailed comments and feedback! We address each of your questions/comments below:
>
> (1) Comparing against prior approaches - Thank you for the suggestion! Please see our general response; in the revision, we have added additional experiments comparing against the BatchTopK SAE (used in [1,2]) and Matryoshka SAE (used in [2,4]). We note that the work [3] also uses the TopK SAE for their results (which is what we use for our main comparisons). Our results indicate that these prior SAE variants all suffer from relatively poor modality alignment, while our GSAE and MGSAE models consistently show improvements on zero-shot classification tasks.
>
> Questions:
>
> (1) Yes - all the baseline comparisons in the paper labeled as "SAE" do exactly this. Comparing the "GSAE" and "SAE" results in our paper indicates that the addition of the group-sparse loss in the GSAE yields significant improvements over the "SAE", which doesn't have this loss term and only optimizes for reconstruction error.
>
> (2) This is relatively easy to ensure through proper selection of the $\lambda$ parameter (as is common for any regularization method). In our experiments, we monitor the average sparsity level in each batch during training, and choose $\lambda$ such that this sparsity level doesn't drop significantly below the value of $K=32$, which is enforced by the TopK operation. In our experiments, we have found this to be a useful heuristic for choosing this parameter in a way that encourages multimodal concepts without causing latents to decay too much.
>
> (3) Yes - all the comparisons in the paper between "GSAE" and "MGSAE" can be used to see the effect of adding masking. Overall, our experiments show that random masking improves performance with respect to MMS and dead neuron counts in almost all cases and can sometimes lead to small improvements in zero-shot performance. Through our additional experiments in the Appendix, this is further validated to be the case across different hyperparameter choices and encoders.
>
> (4) Please see above - we have added comparisons against most methods which have previously been applied to the CLIP embedding space, and we find that all of them suffer from relatively poor modality alignment compared to the GSAE and MGSAE we propose
>
> (5) We do not necessarily claim that audio poses unique challenges. Rather, we note that is not a priori clear that SAE methods which have been applied with some success to CLIP embeddings would also work well for other multimodal domains like audio/text embedding spaces. Hence, our results can be viewed as evidence that, with proper training techniques, SAEs can also yield useful concept-based decompositions in the audio/text setting -- a setting in which this has not previously been demonstrated. The work [5] applies SAEs to audio embeddings from Titanet, rather than multimodal audio/text embeddings. Here, our focus is not just on whether SAEs can be applied to audio, but rather on how to make SAEs with *aligned* sparse codes across modalities, where one of the modalities may be audio.
>
> (6) The form of random masking that we propose encourages the decoder to be able to reconstruct paired embeddings from sparse codes with the same support. Intuitively, this encourages inputs of different modalities, but with the same semantic information to activate the same neurons in the autoencoder, leading to more multimodal neurons in the MGSAE, compared to the GSAE. Per our understanding, the fact that this masking is randomized during training also discourages the autoencoder from relying to heavily on some neurons instead of others, leading to a smaller number of dead neurons overall. A more detailed investigation of how the masking affects dead neurons during training would be an intriguing area for future work!
>
> (7) Thanks for the suggestion! We have added a suite of evaluations with a SIGLIP2 backbone to the revised pdf, showing similar overall trends to those we saw for CLIP.

---

### Official Review · Reviewer_Nk7i · 2025-11-01

**Soundness:** 3
**Presentation:** 3
**Contribution:** 2
**Rating:** 6
**Confidence:** 3

**Summary:**

This work studies interpretability of multimodal embeddings using sparse auto-encoder. The main objective is to find aligned sparse multimodal latent vector from sparse autoencoder. To this end, a definition of modality-split dictionary is provided and the author propose multimodal monosemanticity score (MMS). Moreover, with masking approach, their proposed SAE approach shows improved multimodal dictionary from experiments.

**Strengths:**

In terms of interpretability, understanding aligned multimodal embedding is important. The paper propose important concept of modality split dictionary, which I think a crucial for interpretability of multimodal embeddings. Proposed approach is simple yet effective as the experimental results verify the authors claim. Overall, although I think the contribution seems a bit incremental, problem and the proposed definition are worth looking at.

**Weaknesses:**

While I enjoyed reading the paper, I think the paper could be more improved if more experimental analysis is conducted. For example, I do not see ablation study in choosing $K$ in TopK step, and there is no intuitive explanation or implication from it. Moreover, there is no ablation study for $p$ for random mask. It is even difficult to see why the random masking is required. Thus, I think the paper needs more evidence and experiments, which would make it more convincing.

**Questions:**

1. Can the author provide what is the impact on choosing $K$ in interpretability and downstream tasks (e.g., classification)?
2. What is the rationale to use random mask? I saw the random mask gives very marginal gain. Why does this method give such a small gain?
3. Section 5.3 is interesting. Could the author provide more example or similar studies?

---

> ### Author Response · Authors · 2025-11-25
> **Response to Reviewer Nk7i**
>
> Thank you for your detailed comments and positive feedback! We address each of your questions/comments below:
>
> (1) More experimental analysis -  Thank you for the feedback! Please see our general response and the range of new experimental results we have added in the revision. Specifically, we have added extended results applying our method to different embedding spaces (SIGLIP2, AIMv2), comparing to new baselines (Matryoshka SAE, BatchTopK SAE), and across varying hyperparameter choices for expansion ratio and sparsity level K. You are correct that in a few cases, the MGSAE and GSAE appear to have similar performance; however, we believe that viewed as a whole and especially with our newly added results, our experiments show that there is value in random masking, since it improves performance with respect to MMS and dead neuron counts in almost all cases and can sometimes lead to small improvements in zero-shot performance. In either case, we emphasize that both of these proposed methods always significantly outperform standard SAEs which have been considered in the literature.
>
> Questions
>
> (1) Effect of choosing K: Thanks for the interesting question - to address this, we have added new experiments in the Appendix investigating the effect of the expansion ratio and value of K in zero-shot classification. Our results indicate that increasing the choice of K typically leads to improved zero-shot classification performance, while varying the expansion ratio has a more negligible effect.
>
> (2) Random mask: Please see above for why we believe our results as a whole indicate that random masking is a valuable addition to our methodology. Our results indicate that the group-sparse loss leads to the biggest performance gains, while random masking in most (but not all) cases provides a small boost on top of this (and hence is still generally a valuable addition to the training of multimodal SAEs)
>
> (3) Additional interpretability studies: We are glad you found this interesting! We have prioritized ablations and extensions to different encoders/baselines/hyperparameters in the rebuttal period, but time-permitting, we hope to add another example of spurious correlation detection on the Waterbirds dataset. Thanks for the suggestion!

---

### Official Review · Reviewer_GGM9 · 2025-11-01

**Soundness:** 1
**Presentation:** 2
**Contribution:** 2
**Rating:** 2
**Confidence:** 3

**Summary:**

This paper is a well-placed venture in expanding interpretability for multimodal domains by using Sparse AutoEncoders. The authors tackle the issue of modality splitting in the “features” captured by SAEs that are trained on just reconstruction loss for multimodal models. Key contributions are expanding the metric of monosemanticity of a given feature to include the modality of activations and introducing an updated training recipe to improve alignment between the sparse encodings of different modalities.

**Strengths:**

1. Investigates and quantifies the role of modality alignment in feature discovery using SAEs
2. Introduces a simple but seemingly effective method to improve the discovery of interpretable monosemantic features
3. Expands the application of SAEs to audio/text CLAP models alongside image-text CLIP models
4. Discusses implications on downstream tasks like classification, retrieval, and steering

**Weaknesses:**

1. Only considers a limited range of test models (one for each set of modalities), which limits the evidence for the applicability of the proposed method. In particular, state of the art models like AiMV2 are shown to have different distribution of feature weights between modalities compared to the studied encoder [1], and the applicability of the proposed MGSAE method in such situations is unclear.

2. Ablation studies aren’t presented, in particular, different expansion ratios for the SAE (number of features), and different sparsity measures (K) aren’t explored in the body of the text. This leaves unanswered questions about the applicability of MGSAE and if the method would continue to improve alignment and reduce dead neurons in different settings.

3. Does not fully expand on the distribution of activations / neurons between different modalities. In particular, he authors do not delve into the comparative strength of activations between the modalities, nor the frequency of the activated concepts. A show of how the weight of the features are split between the modalities would be very useful evidence to further justify the paper’s claims of improved alignment between modalities using the presented method.

4. Does not present baselines / benchmarks comparing the introduced MGSAE method with other known works that present potential improvements in multimodal settings over the TopK SAE method. In particular the methods proposed in [2] which are cited by the work would help position this work in the existing literature.

5. While the MMS metric is motivated in intuition by the paper, no quantitive comparisons are present to further validate the measure compared to the results in [3]. An understanding of how MMS and raw MS compare, especially between the known methods and those introduced in the paper could further cement the metric’s usefulness.

[1] “Interpreting the Linear Structure of Vision-language Model Embedding Spaces”, COLM 2025

[2] “Interpreting CLIP with Hierarchical Sparse Autoencoders”, ICML 2025

[3] “Sparse Autoencoders Learn Monosemantic Features in Vision-Language Models“, NeurIPS 2025

**Questions:**

1. In Figure 5, the right side (MGSAE) shows 2 entires of concepts with the name “beautiful blonde” - do the authors understand why this might have happened and how the actual underlying concepts differ ?

2. Table 1: Do the authors any intuition as to why there seem to be cases where the (M)GSAE methods perform so competitively despite seemingly having many dead neurons ? Would a potential No SAE baseline for these tasks be helpful in building understanding?

3. Have the authors considered if the MGSAE method learn stable multimodal features even with random masking ? That is to say, could the concepts learned by the dictionaries in differently seeded runs be aligned ?

Nits:
* Line 176: The ‘W’ is missing the subscript for ‘W_dec`
* Line 641: Switch up between z and x, y

---

> ### Author Response · Authors · 2025-11-25
> **Reviewer GGM9**
>
> Thank you for your detailed comments and helpful feedback. We have included several additional experiments to address your key concerns (see general response), and we address each of your questions/comments below:
>
> (1) "Only considers a limited range of test models"  - Thank you for the suggestion! In the revision, we have expanded our results to evaluate the GSAE and MGSAE on two new encoders, the popular SIGLIP-2 and AIMv2 multimodal encoders. In each case, we see that the MGSAE consistently has the smallest number of dead neurons and the largest number of multimodal neurons. Additionally, the GSAE and MGSAE both consistently outperform the SAE on zero-shot evaluations, indicating improved alignment from using our methodology.
>
> (2) "different expansion ratios ... and different sparsity measures (K) aren’t explored" - Thanks for the suggestion. In the Appendix, we have added additional experiments to investigate the effect of the expansion factor and the choice of K. Specifically, we find that the key insights of our paper are consistent across a wide range of configurations for these hyperparameters. In all configurations, the MGSAE achieves the highest zero-shot classification performance on ImageNet, indicating improved multimodal alignment relative to the baseline SAE.
>
> (3) "Does not fully expand on the distribution of activations / neurons between different modalities" - We believe that our evaluations of dead neurons, zero-shot performance, and MMS provide a comprehensive suite of metrics by which to validate the improved multimodal alignment of our proposed approaches across a variety of multimodal encoders and domains (audio/text and image/text). We also highlight the fact that in most applications of SAEs, the cosine alignment (rather than the exact activation strength of each modality) is most useful metric of the usefulness of a concept dictionary.
>
> (4) Other baselines: In the revision we have added additional experiments with the BatchTopK and Matryoshka SAE models, which are the two main models that have been considered for CLIP embeddings in prior works. Notably, we find that both models have very poor modality alignment, as evidenced by the zero-shot performance on image/text classification tasks. We believe that these new results improve the contextualization of our results within the literature and provide stronger evidence for the utility of our proposed methods.
>
> (5) MS vs MMS: Thanks for the feedback! We have added an expanded discussion on how these metrics vary (along with comparisons to another metric from the literature) in the main text, and we hope that this helps clarify the differences. In addition to applying across modalities, the MMS score applies a normalization scheme to the co-activations which leads a natural interpretation of MMS as a weighted average of cosine similarities. We are also working towards including further numerical comparisons in the remainder of the rebuttal period.
>
> Questions:
>
> (1) "Figure 5" - This is a good observation! This indicates that these two corresponding dictionary elements were most aligned with the same bigram in the vocabulary corpus we used for labeling (and hence are also relatively well-aligned with each other). Interestingly, the multimodal dictionary we construct in the proof of Theorem 1 also has these kinds of aligned dictionary elements - a more detailed study of the geometry of the learned dictionary elements would be an intriguing direction for future work.
>
> (2) We note that, although the (M)GSAE methods have many dead neurons, our results indicate that the number of dead neurons is reduced compared to standard SAEs. Due to this, and particularly because of the increase in neurons that activate for both modalities (cf. Fig. 3), the performance of these methods on zero-shot tasks is improved. For comparison, we include the baseline performance of "No SAEs"  (i.e., the original CLIP/CLAP embedding performance) in the bottom rows of Table 1. Of course, our methods do not typically outperform these dense embeddings, but our main goal is to demonstrate the significant benefit of our methods over standard SAEs in multimodal settings.

---

### Official Review · Reviewer_NjK2 · 2025-11-04

**Soundness:** 3
**Presentation:** 3
**Contribution:** 3
**Rating:** 6
**Confidence:** 4

**Summary:**

The authors propose both a metric and a methodology for better understanding and training sparse autoencoders (SAEs) on vision language model spaces. They propose a metric that combines how useful SAE codes are for aligning semantically aligned pairs, combined with how much SAE codes activate for multiple modalities or just one. They propose that instituting a group-sparse (GS) loss for SAEs is the solution for split-modality concepts, and show that their group-sparse SAEs are better along a suite of metrics like fewer dead codes, more semantic alignment, and better cross-modality transfer.

**Strengths:**

This paper nicely distills some of the recent research on VLM interpretability into a nuanced metric and and interesting new methodology.

I find the evaluations of the GS-SAEs convincing, with both the intrinsic SAE evaluations of dead codes and MMS, as well as the more extrinsic zero-shot transfer (which comes out within a reasonable distance of the solidskyline of the actual embeddings), being interesting.

**Weaknesses:**

I think this is a paper that makes a solid contribution, and I don’t consider any of the below weaknesses to be especially strong.

W1 I don’t feel that the paper has analyses or ablations that increase our intuitions in what it is about Sparse Group loss that causes the decrease in dead neurons, and the multimodal concepts (the latter is more clear). This makes the paper weaker, as what we can mostly get from it is that group-sparse SAEs are better, and so for multimodal models we should switch to them. Since as I understand it the paper would be strongest if it is meant to increase our intuitions about what makes good training for linear concepts, I would like to see another analysis helping with that. Possible questions are: How does the group-sparsity play out in the training dynamics? How do the dictionaries differ geometrically when we include or don’t include that

**Questions:**

Q1 What are the issues that arise from training an SAE on paired data? Might it be a problem that paired data is less broad in domain? Is there a way to integrate these findings about the GS loss with the advantages of using a broader base of data.

Q2 Do you have any idea about why dead neurons are so influenced by the GS loss, and why this seems to be linked also with the multimodality of features?

Q3 It took me some time to understand how the MMS metric differs from some other similar metrics (like the BridgeScore from Papadimitriou et al 2025), so a slightly expanded explanation would be great. After reading through it more carefully, I think I’m convinced that it’s a subtle way of going about measuring split-modality. Could you expand on the metric, and provide some helpful analysis on if there is interference between monosemanticity and multimodality and how this can be quantified?

Q4 It seems that the main methodological contribution of this paper is the group-sparse loss term, but it is not very intuitively explained in the text (eg after line 269). Can you expand slightly on what group-sparsity means and why joint support is encouraged when using this norm? The text mentions this, but as it is the main methodological contribution to the paper I think a slightly more developed theoretical side could be useful.

Q5 (minor) Is alignment targeting the same evaluation as monosemanticity? I’m not sure that aligned pairs is a measure that can be translated trivially to mean monosemanticity, it might warrant a sentence or two in the rewrite if you think so

Q6 (minor) I didn’t find Figure 2 very helpful, it seems to emphasize that there are two encoders/decoders, when in fact the main contribution is the loss term which is in quite small font.

---

> ### Author Response · Authors · 2025-11-25
> **Response to Reviewer NjK2**
>
> Thank you for your detailed comments and positive feedback! We address each of your questions/comments below:
>
> (W1) Thank you for the feedback! Regarding ablations, we have added additional experiments for different encoders (SIGLIP2 and AIMv2), choices of expansion factor, and sparsity level K in the revision. We believe that these additional experiments provide very strong evidence that group-sparse loss improves performance in a wide range of multimodal settings. We don’t explicitly study the training dynamics induced by group-sparsity (though we agree this would be a fascinating area for future work!). Instead, we take the perspective that “good training for linear concepts” is one that induces multimodal dictionaries, and we show that the MGSAE succeeds at learning such a dictionary in many settings, relative to previous approaches, which are quite poor when evaluated from this perspective.
>
> (Q1) Thanks for the question -- it is straightforward to adapt our proposed methodology to settings where large amounts of unpaired data are also available. In such settings, the group-sparse loss can be applied only for paired samples, and reconstruction loss alone can be used for the remaining data samples. While an extensive investigation of this is outside the scope of this work, we believe that using the group-sparse loss with even a small amount of paired data would provide improvements over the baseline SAE in multimodal settings, and we have added a discussion of this in the revised manuscript.
>
> (Q2) The GS loss encourages latents corresponding to paired samples to have the same sparsity pattern. Intuitively, this means that more neurons will co-activate across modalities, leading to fewer dead neurons for each individual modality and more multimodal neurons. One heuristic interpretation is that the group-sparse loss allows for neurons which would normally have zero activation for one modality to become active if they are useful for the other modality.
>
> (Q3) Our MMS metric measures a fundamentally different quantity than the BridgeScore in Papadimitriou et al. 2025. The BridgeScore assigns a value to each pair of neurons/concepts (answering the question of how often neuron i and neuron j both activate across paired modalities), while our MMS score assigns a value to a single neuron/concept (answering the question of how often a single neuron co-activates for semantically similar inputs of different modalities). Unlike Papadimitriou et al. 2025,  we explicitly aim to learn a dictionary where each concept is truly multimodal, so our metric is designed to measure this. Additionally, we measure semanticity by using cosine similarity as a proxy, rather than requiring additional paired data to compute the metric. We agree that some of these differences are subtle, so in the revised pdf we have added additional discussion of this.
>
> (Q4) The use of the group-sparse norm is standard in signal processing problems where it is desirable to encourage a structured sparsity pattern in the learned parameter (see the references in Lines 303-304). Intuitively, consider the matrix of dimension p x 2, containing the sparse codes of data from 2 different modalities. To understand why this loss enforces joint support, note that the group-sparse loss is the L1 norm of the vector containing the Euclidean (L2) norm of each row of this matrix. Due the sparsity-inducing nature of L1 norm regularization, this encourages entire rows of the matrix to be jointly sparse (i.e., a certain concept should activate for either both or neither modality).
>
> (Q5) We agree that coming up with quantitative measures of monosemanticity is a challenging and important area. Cosine similarity of activating inputs is one proxy which has been shown to correlate well (qualitatively) with perceived semanticity in various prior works (e.g., see the visual examples in Pach et al. 2025). We follow (and extend) the precedent established in this literature when developing our evaluation metrics.
>
> (Q6) Thanks for the suggestion - our goal with this figure is to outline the overall methodology, emphasizing the use of paired data (hence two encoders/decoders in the schematic), the positioning of the random mask in the pipeline, and the use of the group-sparse loss. We will be sure to increase the font size for clarity in the finalized revision

---

### Author Response · Authors · 2025-11-25
**General Response to Reviewers**

We sincerely thank all reviewers for their constructive feedback. We are encouraged that the majority of reviewers evaluated the paper positively and that several aspects of our work were consistently highlighted as strengths by reviewers: the effectiveness of the proposed group-sparse autoencoder framework, the utility/flexibility of the proposed Multimodal Monosemanticity Score, the broad and convincing nature of our evaluations, and the novel extension of SAEs to the new audio/text setting.


Based on reviewer feedback, we added multiple new experimental results and discussions that we believe address all major reviewer concerns and significantly strengthen the paper. Below we summarize the major updates and additions:

**(1) Applications to new image/text embedding spaces:** Reviewers GGM9 and G3nW suggested replicating our results for other popular image/text encoders. In Appendix A.3 of the revision, we have added a full suite of experiments for two additional encoders: SIGLIP2 and AIMv2. In both cases, we find that the MGSAE achieves the highest number of multimodal neurons and the smallest number of dead neurons. Moreover, for zero-shot tasks, the GSAE and MGSAE both consistently outperform the vanilla SAE, consistent with our results for CLIP and CLAP embeddings. These results show that our main insights are robust across a wide range of multimodal embedding spaces (3 different popular image/text encoders and 1 audio/text encoder).

**(2) Evaluating the effect of expansion ratio and sparsity level K:** Reviewers (NjK2, GGM9, Nk7i, xuXW) asked how the choice of expansion ratio and sparsity level would affect our results. In the Appendix of the revision, we provide extensive additional experimental results to demonstrate that the key insights in our paper are stable for a wide range of these hyperparameters. Specifically, we compute the zero-shot ImageNet performance of SAE, GSAE, and MGSAE models across a range of choices for these parameters and show that the performance in all cases satisfies MGSAE > GSAE > SAE, indicating that the group-sparse loss and masking scheme we propose consistently improves modality alignment over baseline SAEs.

**(3) Comparisons to other SAE variants:** Responding to suggestions from Reviewers GGM9 and G3nW, we show that the poor performance of standard SAEs is not limited to the TopK SAE architecture. In particular, we add comparisons in the main paper to the BatchTopK and Matryoshka SAEs considered in previous works which study the CLIP embedding space [1-2]. In both cases, the zero-shot performance of sparse codes is poor compared to our proposed GSAE and MGSAE approaches. These additional results further underscore the importance of improving alignment of SAE embeddings in multimodal settings and provide strengthened evidence that the GSAE and MGSAE yield significant advances in this space.

[1] “Interpreting the Linear Structure of Vision-language Model Embedding Spaces”, COLM 2025

[2] “Interpreting CLIP with Hierarchical Sparse Autoencoders”, ICML 2025

**(4) Expanded discussion of how MMS compares to previously proposed metrics:** To clarify questions from Reviewers NjK2 and GGM9, we add additional discussion in the main paper of how our proposed MMS metric differs from the MS metric of Pach et al. 2025 and the BridgeScore metric of Papadimitrou et al. 2025, which were also introduced in the context of multimodal CLIP embeddings.

In addition to these primary changes, we also clarify wording in a few places and address several minor reviewer concerns, which we address in the specific response to each reviewer.

---

### Meta-Review · Area_Chair_QQWt · 2026-01-03

**Summary:**

The reviewers generally agreed that the paper addresses an important problem in VLM interpretability: the tendency of Sparse Autoencoders (SAEs) to learn "split dictionaries" (features that are active only in one modality) when applied to multimodal spaces like CLIP.

The initial reviews had two conflicting perspectives:
* The use of group-sparse regularization ($\ell_{2,1}$ norm) and cross-modal masking to enforce alignment was positively received by the reviewers for their theoretical elegance, conceptual reasonableness, and implementation simplicity.
* All four reviewers raised concerns on insufficient experiments and ablations. Reviewers `GGM9` and `G3nW` expressed concerns regarding whether these findings were specific to the TopK SAE architecture or the original CLIP encoders.

The recommendation for Acceptance is based on the authors' rebuttal that adequately addressed the reviewers' concerns with additional experiments.

**Reviewer Concerns:**

Concerns Addressed by the Rebuttal

* Generality across encoders (`GGM9`, `G3nW`, `xuXW`): Reviewers were concerned that results were limited to CLIP. The authors added additional results for SIGLIP2 and AIMv2. The results consistently showed that their Masked Group-Sparse Autoencoder (MGSAE) achieved the highest multimodal neuron count and best zero-shot performance across all four backbones.

* Comparison to prior art (`GGM9`, `G3nW`): Reviewers noted the absence of comparisons to other multimodal SAE attempts. The authors added comparisons to BatchTopK SAE and Matryoshka SAE. These experiments confirmed that MGSAE provides superior modality alignment compared to existing alternatives in the literature.

* Hyperparameter sensitivity (`NjK2`, `Nk7i`, `xuXW`): Concerns regarding the expansion ratio and sparsity level ($K$) were resolved through additional sensitivity plots in the Appendix. The authors demonstrated that the performance ranking (MGSAE > GSAE > SAE) remains stable across diverse configurations.

* Metric clarity (`NjK2`, `GGM9`): The authors provided a detailed differentiator between their MMS metric and previously proposed metrics like BridgeScore, clarifying that MMS uniquely measures single-neuron co-activation for semantically similar inputs.

Outstanding Concerns

* Convergence vs. existence (`xuXW`): While Theorem 1 proves that a multimodal dictionary can exist, there remains a theoretical gap in characterizing why stochastic gradient descent (SGD) successfully converges to this specific solution in a non-convex landscape.

* Diversity vs. alignment trade-off (xuXW): There is still limited analysis on whether forcing multimodal alignment inadvertently suppresses modality-unique features (e.g., specific visual textures that have no linguistic equivalent).

**Reviewer Scores:**

The authors' rebuttal have added extensive new results, addressing most of the reviewers' initial concerns.

* Reviewer `NjK2` (Initial: 6 $\rightarrow$ Estimated: 7): Their concerns were primarily about intuition. The authors' explanation of how joint support reduces dead neurons through co-activation was convincing.

* Reviewer `GGM9` (Initial: 2 $\rightarrow$ Estimated: 6): This reviewer gave a very low score based on the "limited range of test models" and insufficient experiments/ablations. Given that the authors added exactly the models (SIGLIP2, AIMv2) and baselines (BatchTopK) requested, as well as various ablation results as summarized in the general response, the empirical basis for the score of 2 has been effectively dismantled.

* Reviewer `Nk7i` (Initial: 6 $\rightarrow$ Estimated: 6): The $K$-sparsity ablations provided in the Appendix satisfy the reviewer's request for "more evidence."

* Reviewer `G3nW` (Initial: 4 $\rightarrow$ Estimated: 6): The inclusion of Matryoshka SAE baselines and SIGLIP2 backbones resolves the "limited comparison" weakness that drove the initial below-threshold rating.

* Reviewer `xuXW` (Initial: 6 $\rightarrow$ Estimated: 6/7): The authors provided the requested qualitative "violin" steering example and a mathematical framework for scaling to $n$ modalities, strengthening an already positive review.

---

### Decision · Program_Chairs · 2026-01-26

Accept (Poster)